# Mammalian Models in Alzheimer’s Research: An Update

**DOI:** 10.3390/cells12202459

**Published:** 2023-10-16

**Authors:** Himadri Sharma, Keun-A Chang, John Hulme, Seong Soo A. An

**Affiliations:** 1Department of Bionano Technology, Gachon Bionano Research Institute, Gachon University, 1342 Seongnam-daero, Sujeong-gu, Seongnam-si 461-701, Gyeonggi-do, Republic of Korea; 2Neuroscience Research Institute, Gachon University, Incheon 21565, Republic of Korea

**Keywords:** Alzheimer’s Disease, mouse models, non-human primate, β-amyloid, tau

## Abstract

A form of dementia distinct from healthy cognitive aging, Alzheimer’s disease (AD) is a complex multi-stage disease that currently afflicts over 50 million people worldwide. Unfortunately, previous therapeutic strategies developed from murine models emulating different aspects of AD pathogenesis were limited. Consequently, researchers are now developing models that express several aspects of pathogenesis that better reflect the clinical situation in humans. As such, this review seeks to provide insight regarding current applications of mammalian models in AD research by addressing recent developments and characterizations of prominent transgenic models and their contributions to pathogenesis as well as discuss the advantages, limitations, and application of emerging models that better capture genetic heterogeneity and mixed pathologies observed in the clinical situation.

## 1. Introduction

Accounting for nearly 70% of all dementia cases with an average course of ten years, AD has the potential to overwhelm healthcare systems generating large care deficits. Unfortunately, such deficits are already prevalent in many healthcare systems requiring individuals to compensate (financial or otherwise) for the shortfall. For example, in 2017 and 2020, 16 and 11 million USA citizens provided 18.5 and 14 million (family members and health workers) hours of unpaid care, respectively [1]. However, even with continued contributions, by 2050 the USA’s AD care model will become untenable as predicted costs surpass $1 trillion/annum. In addition to economic concerns, the negative impacts on other health sectors (mental) [2] are yet to be fully realized, with burnout and depression frequently reported among carers. Thus, the current directives for AD scientists and clinicians have shifted from developing end-stage treatments to ones that extend the early stages of AD, encouraging arresting actions and deficit reductions among patients and carers alike.

AD can present as early (<65 years) onset Alzheimer’s disease (EOAD) or late (>65 years) onset Alzheimer’s disease (LOAD). The disease itself has several stages and is characterized by three pathological hallmarks, extracellular amyloid beta (Aβ) [3,4,5] plaques (dense and diffuse), intracellular neurofibrillary tangles (NFT), and cortical and hippocampal neurodegeneration [6,7,8,9,10]. The formation of extracellular Aβ plaques starts with the sequential cleavage of amyloid precursor protein (APP) by a β-secretase, β-APP cleaving enzyme (BACE1), an aspartyl protease followed by γ-secretase cleavage (a protein complex composed of presenilin-1, nicastrin, APH-1, and PEN-2) [11,12,13,14,15,16,17] leading to the release of three protein fragments including Aβ peptides (Aβ40, Aβ42, extracellular space), the amyloid-intracellular domain (AICD) (cytoplasm), and soluble APP. 

For the most part, the majority of Aβ peptides produced by γ-secretase end at amino acid 40 (Aβ40) and a minority end at amino acid 42 (Aβ42). However, it was not until the discovery of familial mutations (1991) in the amyloid precursor protein (APP) and presenilin (PSEN1 and PSEN2) genes resulting in APP cleavage favoring the aggregation of amyloid-beta 42 (Aβ42), that the amyloid the cascade theory was proposed. In short, the initial theory followed a simple process, β-amyloid deposition → tau phosphorylation and tangle formation → neuronal death. Since then [18], more than 300 mutations in APP, PSEN1, and PSEN2 genes have been identified [19]. Interestingly, the age of onset (AAO) and the severity of symptoms are markedly different between mutations. For example, carriers of pathogenic PSEN1 variants may present symptoms before 25 or after 60 years of age [20]. Moreover, numerous findings have emerged, suggesting tangle formations can precede amyloid deposition, and the degree of cognitive impairment correlates much better with NFT counts than amyloid plaque burden [21] challenging the original theory.

NFTs are composed of aggregated tau proteins, collectively called tauopathies, a major pathological feature of AD and other neurodegenerative conditions. Under normal conditions (homeostasis), tau is an unstructured axonal protein that binds and stabilizes microtubules [22]. The protein can present as a three-repeat (3R) or four-repeat (4R) isoform as exon 10 on the microtubule-associated protein tau gene (MAPT) is subject to alternative splicing [23]. The resultant isoforms can undergo oligomerization and hyperphosphorylation, contributing to the formation of intracellular NFT in the axons [24,25]. Aβ and tau accumulation in the brain promote several pathophysiological (breakdown of the myelin sheath around neurons, astrocytic and microglial inflammation) changes affecting AD patients’ cognitive function. In addition, Aβ and tau accumulation remain central to the histopathological post-mortem staging (Braak) systems, considered the gold standard of AD diagnosis. 

These genetic, biochemical, and neuropathological hallmarks warrant the creation and validation of AD animal models. Beyond the dominant forms of AD, mutations in apolipoprotein E (APOE) and the triggering receptor expressed on myeloid cells 2 (TREM2) are the most common genetic risk factors for LOAD. 

Preclinical investigations using murine models are essential to the arsenal of tools to probe specific disease mechanisms and evaluate therapeutic efficacy and diagnostic strategies within a living organism. Of the 214 murine models registered between 1988–2022, 212 were transgenic, one was naturally occurring, and one was chromosomally induced [26]. When selecting a model from the register, matching the murine model (laboratory strain) to the experimental question (for example, stage of the disease) under study remains the primary consideration [27]. The current plethora of available Tg models permits researchers to explore up to six pathological hallmarks, including plague and tangle formations, neuronal loss, synaptic loss, gliosis, cognitive impairment, and changes in long-term potentiation/long-term depression (LTP/LTD), but generally not all in the same model. In this review, the advantages and limitations of prominent (based on the amyloid cascade hypothesis) Tg and non-transgenic models engineered to emulate the major pathological hallmarks of EOAD, then LOAD, are assessed. In addition, diagnostics (behavior, biomarkers, neuroimaging) used to evaluate the therapeutic potential of a selection of models are discussed (Figure 1).

## 2. Murine Models of Amyloid Pathology

### 2.1. PDAPP

The first amyloid model was presented in 1995 [28], expressing high levels of human APP with the V717F Indiana mutation. These transgenic mice progressively developed pathological hallmarks of AD as the mutation increases the ratio of Aβ42/Aβ40. These Tg mice exhibited various features resembling human AD-like extracellular amyloid fibrils organized in plaques, degenerative subcellular changes, dystrophic neuritis, apoptosis, gliosis, and synaptic loss that spread progressively from the hippocampus to the cortex [28,29]. These mice also presented age-related memory loss (with highest at 12–15 months of age) which is the main feature in Alzheimer’s patients [30,31,32]. Aβ plaques were observed at 6 months with no tau tangles. Synaptic loss presents after 8–9 months, whereas no neuronal loss is seen. Cognitive impairment was seen by 3 months and gliosis was seen by 6 months [27].

### 2.2. Tg2576

In 1996 Karen Hsiao and colleagues developed the Tg2576 transgenic mouse. The model carries a double Swedish mutation K670N and M671L and is synonymous with early onset familial AD [27,33]. Utilizing a prion promoter (PrP) that non-specifically drives the hAPP gene, the model demonstrated up to a 5-fold increase in APP production, with amyloid pooling in the heart, kidney, and lungs in the first 3 months. Synaptic loss was observed within 4–5 months [34,35]. This was followed by reduced cerebral blood flow by 8 months, spatial working memory deficits by 5 months, and CAA by 9–10 months. Plaque formation was observed at 11–13 months in the temporal, frontal, entorhinal cortices, hippocampus, pre-subiculum, and cerebellum. No profound cognitive impairment was seen [36], even though these mice can sometimes display hyperphosphorylated tau in the last months of life [27]. 

### 2.3. APP23 

Sturchler-Pierrat et al. [37] described the APP23 mouse in 1997. Akin to the previous model, these mouse models harbored the Swedish double mutation on the hAPP gene and utilized a neuronal promoter instead of a prion promoter to induce hAPP overexpression [37]. As a result, soluble Aβ blood levels remained stable in the APP23 mice over time [38], presenting Aβ plaques and spatial memory deficits earlier than in the Tg2576 mice at 3–6 months. Interestingly [39], when the APP23 APOE allele was silenced, vascular damage and oxidative stress occurred; conversely, APOE4 overexpression in Tg2576 mice induced oxidative stress and vascular injury [40]. 

At six months, vascular abnormalities were observable in APP23, with integrin’s overexpression on the platelets’ surface, resulting in collagen and fibrinogen adhesion and vessel occlusion. These were accompanied by elevated IgG levels (1.2-fold) in the neurons, blood vessels, and glial cells, which were absent in age-matched wild-type mice, implying that enhanced IgG synthesis may be unique to this model. A reduction in cerebral blood flow at 11 months was observed, with microhemorrhage and CAA present at around 18 months [41]. At the same time, other vascular irregularities, such as twisted or studded blood vessels with protrusions and reduced micro-vascular flow in the frontal and temporal cortices in APP23 mice, were evident [40]. Multiple studies indicated CAA and cerebral blood flow changes precede vascular injury [38,42]. Therefore, this model may be considered suitable for cerebral vascular investigations.

### 2.4. APP(V717I) & J20 Models

Harboring a common familial APP mutation (London V717I) this 1999 model also utilized a neuronal Thy1 promoter to drive hAPP overexpression which lead to spatial memory deficits by 6 months, Aβ plaque formations by 10 months, CAA by 15 months, and micro hemorrhages by 25–30 months [42,43]. However, a recent clinical report [44] regarding the neuropathologic characterization of an APP V717I carrier, demonstrated that genetically determined ADAD can show significant phenotypic variability including widespread parenchymal beta-amyloid (Aβ) deposition, severe cerebral amyloid angiopathy (CAA), and predominant age-related TDP-43 proteinopathy (LATE). 

In 2000, the J20 model carrying a hAPP gene with three mutations (Indiana (V717F) & Swedish double-mutation (K670N and M671L)) and a PDGF-β promoter [45] was introduced. J20 exhibited elevated soluble Aβ brain levels after 1 month, neuronal loss (3 months), spatial memory impairment (4 months), and Aβ1-42 plaque morphologies (medium and diffuse) at 5–7 months [46]. Micro hemorrhages, CAA, and reduced flow ensue at 11 months, interestingly [47,48], and J20-hAPP mice have recently shown enhanced baseline blood volumes and saturation of all vascular compartments under hyperoxic conditions, suggesting that it could be used to study potential treatments or early preventive actions, such as saturated oxygen or moderate hypoxic conditioning.

### 2.5. TgCRND8 Model 

Akin to J20, the TgCRN8 model also harbors’ the hAPP gene with the Swedish and Indiana mutations utilizing a prion promoter instead of PDGF-β, resulting in hAPP overexpression in neuronal tissue [49,50]. TgCRND8 mice exhibited increases in the Aβ42/Aβ40 ratio, leading to an accumulation of Aβ amyloid plaques, intraneuronal Aβ, soluble oligomeric Aβ (oAβ), and insoluble fibrillar Aβ (fAβ). The TgCRND8 mice developed spatial deficits at 3 months, followed by Aβ plaques at 5 months, significant (>30%) neuronal loss at 6 months, and tauopathies between 7–12 months [51]. CAA and reduced cerebral blood flow developed later in the J20 mice than in the TgCRND8 mice (11 months vs. 6 months) due to impaired Aβ clearance [52]. From 12 months onwards, the TgCRND8 mice developed capillary CAA. The TgCRND8 mice exhibited a wide spectrum of result, including Aβ plaques, fibrils, and oligomers exhibiting greater neuronal toxicity than other models (Dutch (E693Q), that solely favor the overproduction of intraneuronal oligomers.

### 2.6. TgSwDI Model

Introduced in 2004 [53], the TgSwDI mouse AD model harbors an hAPP gene segment with three mutations Swedish (Lys670→Asn/Met671→Leu), Dutch ((E693Q)Glu693→Gln), Iowa (D694N (Asp694→Asn)), and the promoter Thy1.2 [53]. Tg-SwDI mice express transgenic human AβPP only in the brain, but at levels below those of endogenous mouse AβPP. Moreover, the model showed that the overexpression of human AβPP is not necessary for the development of Aβ pathology in the mouse brain. Another unique characteristic of this model is the accumulation of numerous diffuse nonfibrillar plaques and fibrillar deposits in the parenchymal tissue. In TgSwDI mice, reduced cerebral blood flow, and memory deficits appeared as early as 2–3 months, resulting from a robust accumulation of insoluble Aβ40 and Aβ42 [54]. In addition, further investigations suggest that said mutations drive Aβ deposition in the cerebrovasculature, leading to type 1 CAA and vascular fragility with cerebral hemorrhages at 6 months [55], followed by extensive deposition at 1 year. 

Primarily fibrillar, the deposition is closely related with a severe inflammatory reaction, knocking out the APOE allele and causing a shift from microvascular to parenchymal Aβ deposits without changing the overall Aβ burden, indicating that APOE is the driving force for CAA and microhemorrhages in this model [42]. Moreover, [56] a longitudinal study demonstrated that APOE protein levels increased 3-fold in 9-month-old TgSwDI mice compared to 3-month-old wild-type mice, indicating that CAA may also increase with age in these mice, suggesting that TgSwDI is well suited to therapeutic investigations regarding Aβ degradation and clearance.

### 2.7. 5xFAD-

The 5xFAD double transgenic mouse model was developed in 2006, containing the following 5 familial AD mutations: APP KM670/671NL (Swedish), APP I716V (Florida), APP V717I (London), PSEN1 M146L, and PSEN1 L286V [57,58]. The 5xFAD model shows an aggressive amyloid deposition accompanied by gliosis, synaptic, and neuronal loss [59]. β- and γ- secretase are the two proteases that are responsible for cleaving Aβ from APP. APP is cut at the N terminus of Aβ domain by β- secretase to produce C99 (membrane bound fragment) and secreted APP ectodomain APPsβ. C99 is then cleaved by γ- secretase to generate the C terminus of Aβ. Since the γ- secretase cleaving is not precise, Aβ peptide of 38–43 amino acid length are produced. A higher proportion of Aβ40 (40 amino acid) is produced under normal condition [57]. Intraneuronal Aβ depositions developed within 1.5–2 months of age, with memory deficits at 4 months, and neuronal loss at nine months [57] accompanied by cognitive and motor deficiencies, with observable tau tangles [60]. In addition, amyloid pathology in the spinal cord was observed at around 11 weeks in the lumbar regions and cervical extending along the cord by 19 weeks [61]. According to the study done by Oakley et al. in 2006, intraneuronal accumulation of Aβ occurs mainly in the neurons present in the deep cortical layers and subiculum in 5XFAD mice model before the deposition of amyloid. Some plaques appeared to be originating from the intraneuronal Aβ-containing neurons that have degenerated morphologies [57]. All the above features of the 5xFAD mice make it a popular choice for AD research.

### 2.8. APPxPS1-

The APPxPS1 transgenic mouse model contains the following mutations: APP K670_M671delinsNL (Swedish) and PSEN1 L166P. This model shows a 3-fold higher expression of hAPP transgene than murine APP. Plaque deposition occurred in 1.5 months in the cortex and in 3–4 months in the hippocampus [62]. Gliosis occurs at around 1.5 months of age with an increase in astrogliosis. Synaptic loss occurred after almost 4 weeks of plaque formation which continued for several months [63]. Cognitive impairment occurred at around 7 months [64], but was also reported at 8 months by Radde et al. [62]. According to Gengler and colleagues, no detrimental effects on synaptic plasticity was found in young mice up to 5 months of age as plaque formation, Aβ production and inflammatory markers are relatively low. Impaired LTP in the hippocampal CA1 region was found between 8–15 months of age [65]. No severe neuron loss was reported. According to Rupp et al., neuron loss occurred at 17 months of age in the dentate gyrus and other subregions with high neuronal density [66]. 

### 2.9. APP NL-F KI-

This transgenic mouse model was first reported in 2014 by Saito et al. and contained the following mutations: APP K670_M671delinsNL (Swedish) and APP I716F (Iberian). APP NL-F mice recapitulate AD pathologies like Aβ pathology, neuroinflammation, and memory impairment, all occurring age-dependently [67]. Amyloid plaques occurred at 6 months in the hippocampus and cortex region. Gliosis also occurred at around the age of 6 months, concurrent with plaque formation. Synaptic loss occurred at 9–12 months, memory impairment was seen at 18 months, and no tangle formation and no loss of neurons was observed [26,67]. The Swedish mutation elevates the level of Aβ_40_ and Aβ_42,_ whereas the Iberian mutation is responsible for increasing the ratio of Aβ_42_ to Aβ_40_ [67]. Two other related strains containing the mutant APP with the Swedish mutation alone (APPNL) or with Iberian, Arctic, and Swedish mutation (APPNL-G-F) were developed. APPNL showed a less severe phenotype than APPNL-F, whereas APP^NL-G-F^ showed a more severe phenotype, including aggressive Aβ deposition [26]. APPNL-F mice are a good candidate to study the effects of elevated Aβ against wild-type levels of APP.

### 2.10. TREM2- BAC × 5XFAD-

The TREM2 receptor is expressed by microglia in the brain and mediating microglial phagocytosis and inflammatory stimuli [68]. Several mouse models are available to study the effects of TREM2 in AD pathogenesis, including TREM2-BAC × 5XFAD [69], TREM2 Humanized (common variant) × 5XFAD [70], and TREM2 Humanized (R47H) × 5XFAD [2]. The TREM2-BAC × 5xFAD model produces less cortical amyloid in 7 months than the 5xFAD model. No data on tau tangles, synaptic, and neuronal loss has been reported. Gliosis has been observed at seven months of age [2]. The R47H variant results in a 3-fold greater risk of developing AD. Mice carrying the human TREM2 gene were crossed with 5XFAD to study the effects of the R47H variant. In a recent study by Song et al. (2018), two transgenic mouse lines were generated, one with a common variant (CV), and another with the R47H variant in TREM2-deficient mice and crossed with the 5XFAD mouse model. The result showed that CV-enhanced microglial activation and augmented plaque-associated microgliosis, indicating that R47H is a loss of function. The presence of soluble TREM2 on neuron cell bodies, on Aβ plaques, and throughout the brain tissue was found in CV mice, but not in R47H variant mice. The study provided evidence of in vivo loss of function in R47H, implying the protective role of TREM2 [70]. A selection of prominent models utilized in Aβ research is shown in Appendix A. 

## 3. Prominent Tau Mice Models

Abnormal hyperphosphorylation of tau protein (p-tau) is a key component of NFT, especially the phosphorylation of specific Thr residues (Thr 181, 217, and 231) is a biomarker of AD pathology [71]. The tau protein can be either 3R or 4R, depending on the presence of the repeat domain. Both of these forms of tau are expressed equally in the adult brain but the 3R/4R ratio gets altered in most tauopathies [72].

The first model to showcase tauopathy expressed 4R (consisting of a group of neurodegenerative diseases which show cytoplasmic inclusions composed of tau protein with the four microtubule-binding domains), the largest tau isoform and the most natural substrate for hyperphosphorylation but showed little-to-no NFT formation [73,74]. Transgenic mice that developed robust neuronal tauopathy were primarily based on transgenic overexpression of mutations that cause FTD with Parkinsonism linked to chromosome 17 (FTD-MAPT) [75,76,77]. 

### 3.1. JNPL3 (P301L)-

Transgenic mice (JNPL3) expressing 4R tau with P301L missense mutation (Pro301→Leu) is a model of neurofibrillary and axonal deterioration, neuronal loss, behavioral deviations, and motor dysfunction [78,79]. Neurofibrillary tangles develop between 4.5–6.5 months of an age, or in a dose-dependent manner. Neuronal loss is observed from 9–10 months, and astrogliosis presents in the diencephalon, brainstem, and basal telencephalon after 10 months [79]. 

To evaluate the effect of P301L, Hutton et al. (2001) developed the Tau Microinjected Model at the Mayo Clinic (Jacksonville, FL, USA). Central to the model is a mutant form of the microtubule-associated protein tau (MAPT) gene linked to chromosome 17 (FTDP-17), which encodes the tau, resulting in protein dysfunction and contributing to neurodegeneration [79,80]. 

The model was developed by microinjecting transgenic constructs containing the P301L mutation and mouse prion promotor into fertilized eggs of hybrid C57BL/6 × DBA2 x SW mice. In this regard, models can be homozygous (002508-M), hemizygous (001638-T), or the wild type (001638-W). Both homozygous and hemizygous microinjected mice displayed a late righting reflex (RR), and their motor functions were feeble within two weeks of the onset of signs [81]. 

### 3.2. hTau-

These mouse models were developed by crossing 8c mice (mice that express tau transgene derived from human PAC, H1 haplotype) [82] with tau exon 1 knockout mice [83,84]. As this model express all the isoforms of human tau, it is the best model to test the effect of drugs on human tau without murine tau intervention [83].

Hyperphosphorylated tau accumulated at 6 months and increased by 13–15 months, with maximum accumulation in the hippocampal and neocortex compared to the brain stem and spinal cord [84,85]. The hTau mice developed cognitive deficits and physiological impairment at 12 months and significant neuronal death occurred after 14 months [86]. Early hyperphosphorylation of tau at Thr (181 and 231) and Ser (202 and 396) residues has been reported in the htau mice model with C57B1/6J strain. Hence, to analyze the efficacy of compounds on tauopathy, the above model and strain can be used at a young age [87,88]. 

### 3.3. rTg (tauP301L)-

The rTg(tauP301L)4510 mouse model expresses the P301L mutation in tau (4R0N) associated with frontotemporal dementia and parkinsonism linked to chromosome 17 [89]. Since its initiation in 2005 [84,85], this mouse model has become popular, as it shows the phenotype of tau pathology. Pronounced neurodegeneration is observed in human tauopathies and provides researchers temporal control over mutant tau transgene expression. For the rTg4510 mice (‘r’ for regulatable), mutant tau was expressed approximately 13times higher than endogenous mouse tau. The degree of neurodegeneration and NFT pathology was aggressive [6,84]. Gender difference has also been reported in this model, with females showing more tau pathology and behavioral deficits than males [90,91]. Accumulation of tau pathology in the form of argyrophilic tangles occurs by 4 months in the cortex and by 5 months in the hippocampus [84,85]. This model develops neuronal loss. For example, Ramsden et al. (2005) and SantaCruz et al. (2005) reported 60% neuron loss in the hippocampal CA1 region at 5.5 months [84,85], whereas Helboe et al. (2017) reported 43% neuron loss between 8–12 months [92]. Gliosis was observed at 2.5 months of age [92], synaptic loss in the dendritic spine at 8–9 months, motor impairments normal up to 6 months, and spatial memory retention impaired in 2.5–4 months [26]. Overexpression of tau_P301L_ in the forebrain of these mice is accepted as the cause of forebrain atrophy and other tauopathy phenotypes, a hypothesis stating that suppression of tau_P301L_ expression by DOX (doxycycline) halted neuronal loss and improved memory function [85]. Mice expressing wild-type human Tau at levels equivalent to tau_P301L_ in rTg4510 mice did not show the development of memory deficits or overt atrophy, showing that the tau_P301L_ mutant form of tau is the direct cause of these phenotypes in rTg4510 [93,94].

### 3.4. PS19 (tau P301S)-

The PS19 mouse model was first developed and reported in 2007 by Lee and colleagues [95]. This widely used tau pathology model expresses the P301S mutant form of human MAPT. Cognitive decline and neurofibrillary tangles formation were observed at six months [26], with an accumulation of the latter in the spinal cord, brain stem, hippocampus, neocortex, and amygdala. Synaptic loss and gliosis were observed at 3 months. Gliosis was observed mainly in the white matter of brain and spinal cord. Neuronal loss was observed at 9–12 months in the hippocampus and cortex, as well as severe neuronal loss by 12 months in the amygdala and neocortex [26,95]. Appendix A summarizes some of the most popular Tau models currently in use.

### 3.5. 3xTg A Composite Model-

3xTg is a triple-transgenic mouse model developed in the year 2003 to demonstrate both Aβ and tau pathology. It contains mutagenic forms of the human MAPT gene segment (P301L), APP695 (KM670/671NL), and human presenilin-1 gene (M146V) all of which are controlled by the neuronal Thy1.2 promoter [96]. In the 3xTg model, Aβ plaques and tau pathology development starts at 6 months and mimics its development in the AD patient. The Aβ accumulates in the cortex and progresses to the hippocampus with age, while tau pathology follows an inverse trend. Additionally, neurogenesis and cognitive dysfunction defects appear within 4 months [97] and gliosis after seven months with no neuronal loss [2,89]. Further investigations showed that neutrophils infiltrate the brain in 4–6-month-old 3xTg-AD mice, contributing to early cognitive decline. Aβ pathology is preceded by tau pathology with NFT at 12 months [27]. The age at CAA onset and microhemorrhage extent are unknown for 3xTg-AD mice. Previous studies found an increased expression of RAGE (Receptors Advanced glycation end products) in 3xTg-AD transgenic mice [98], which in turn induced oxidative stress and expression of proinflammatory cytokines through various pathways [99].

Compared to the C57BL/6 model, 3xTg models show accelerated aging, mitochondrial function, and stem cell composition in the cortex and hippocampus region of the brain [100]. As the 3xTg model exhibits both of the important features of AD, it will be useful in pre-clinical studies to evaluate the effectiveness of compounds in ameliorating AD pathology. Figure 2 shows AD mouse models of amyloid and tau pathology based on their phenotypic characterization.

## 4. LOAD Murine Models

Apolipoprotein E (APOE) (299 amino acids) is highly expressed by astrocytes, followed by microglial cells and neurons in the brain [8,9] and liver. There are three common APOE genetic variants, ε2, ε3, and ε4, with (APOE) ε4 posing the greatest risk factor for LOAD [101,102]. Unlike animals, humans are the only known species that express multiple forms of apoE protein. Only 14% of the human population carries the apoE ε4 allele, and only 50% of AD patients are carriers [103]. Patients with a homozygous ε4 (*APOE*4/4) have a 30–55% risk of developing mild cognitive impairment (MCI) or AD by 85, compared to a 10–15% for those with the *APOE* 3/3 genotype. In other mammals, a similar version of human APOE ε4 protein is expressed [104]. ApoE ε4 leads to the development of AD via various mechanisms, including increased aggregation and decreased amyloid-βpolypeptide clearance; increased tau phosphorylation; reduced glucose metabolism, vascular and mitochondrial dysfunctions; network excitability; neurodevelopmental differences [105,106]. Very few mouse models have been studied for age-related alterations in AD, as existing models, along with behavioral experiments, have failed to show efficacy in clinical trials [107,108]. Studies related to genetics and the biology of LOAD have revealed that other pathways may contribute to disease pathogenesis [109].

Studies in relation to the effects of metal toxicity in Alzheimer’s have been consistently researched, various animal studies included. Copper in trace amounts is essential for the proper functioning of the Central Nervous System (CNS) [110], whereas increase in concentration leads to an increase in free radical generation and oxidative stress [111] as well as enhancement of neuroinflammation and Aβ production [112]. Chronic copper intoxication has been reported to be associated with AD pathology, i.e., accumulation of β-amyloid, neurodegeneration, and cognitive impairment [111,112]. APP possesses Cu binding domains, one in the N-terminal and the other in the Aβ sequence [112,113,114]. White et al. in 1999 reported that the APP gene played an important role in copper metabolism in the liver and brain of APP^−/−^ and APLP^−/−^ (Amyloid Precursor-like protein 2) [115]. In another study, APP^sw/o^ mice, that overexpressed human APP transgene, were exposed to Cu for 90 days, resulting in increased neuroinflammation, Aβ accumulation, memory impairment, and increased level of Cu in brain capillaries [111,112]. Different studies have researched AD pathology induced due to Cu toxicity using different mouse models, such as the PS1/APP transgenic mice model that showed senile plaque-like deposition of Aβ after 6 weeks of exposure to the Cu dose [116]; the Kunming strain mice that displayed increased oxidative stress, neuronal apoptosis, increased APP mRNA levels, and cognitive deficits after 8 weeks of Cu exposure [117]; and the Wistar rat, who, after 2 months of Cu exposure, displayed an increased Aβ (1/42)/(1/40) ratio in cortex and hippocampus, increased nitrate level in brain [111,118,119]. 

There are some other neuroinflammation models that have the potential to be used as SAD models [120], for example, the Polyriboinosinic-polyribocytidylic acid (PolyI:C) induced model. PolyI:C is a double-stranded synthetic RNA that induces neuroinflammation. Increased brain cytokine levels can be observed within 3 weeks, which remain throughout life, in these models [121]. Tau hyperphosphorylation is not observed in 3 months but increases as the mouse reaches 6 months of age with spatial recognition memory impairment observed at 20 months compared to wild-type rodents [121]. Overexpression of human p25 can induce AD-like pathology in mice [120,122]. Neuroinflammation occurs first in p25 transgenic mice marking the start of AD-like pathology [123]. Microglial activation has been observed after 4 weeks of p25 overexpression [122], and cognitive impairment at 6 weeks [124]. p25 transgenic mice have also displayed Aβ deposition, tau phosphorylation, and neuroinflammation [123,125]. A polyether toxin, Okadaic acid (OKA), inhibits serine/threonine polyphosphatases 1 and 2A [120,126]. PP2A dysfunction leads to tau hyperphosphorylation [127], hence, the activity is decreased in AD patients [128]. Few studies have reported that OKA induces memory impairment in rats [129,130]. In 1998, after the infusion of OKA for 4 months in rat brains, tau hyperphosphorylation was observed along with apoptotic cell death after 2 weeks and nonfibrillar Aβ deposition in the cortex after 6 weeks infusion, was reported [131]. 

AD progression is slow in humans, with individuals undergoing MCI and SCD before developing definitive AD. Therefore, more animal models targeting MCI and SCD are needed to study the development of early pathological changes in AD, since SAD is the most common form of AD in humans [120].

## 5. Alternative Murine Models

The senescence accelerated mouse (SAM, transgenic) and senescence-accelerated RAT (OXYS) (non-transgenic) models are some of the most popular phenotypic murine models used in AD research [132,133,134,135]. The SAM model represents a series of inbred strains derived from the AKR/J strain, consisting of nine senescence-prone strains (SAMP) and four senescence-resistant strains (SAMR) with the characteristic of accelerated and normal (controlled) ageing. Among SAMP strains, SAMP8 and SAMP10 mice have exhibited numerous deficits/symptoms consistent with AD’s intermediate and later stages, such as learning and memory, emotional disorders, altered circadian rhythm, and specific biochemical, pharmacological, and pathological changes. For example, SAMP8 has shown passive and active avoidance responses indicative of memory impairment and cognitive decline at 2 months, followed by tau hyperphosphorylation at 3 months, inflammation at 5 months, Aβ deposition at 6 months, synaptic degeneration at 8 months, and neuronal loss at 10 months [136]. Moreover, these strains have manifested various pathobiological phenotypes spontaneously, suggesting they may have a significant role in investigating future drug candidates such as J147 and 6-chlorotacrine (huprine)-TPPU hybrids [137,138].

Sharing similar traits with the previous model, OXYS rats can elucidate the mechanisms of ageing and pathology and provide an objective evaluation of potential therapeutics and preventive actions [139]. For example, a recent study using said model assessed the association of cerebrovascular dysfunction with the development of AD-like pathology. As a result, the maximal density of blood vessels in prepubescent mice was observed on day 20, followed by a reduction in the vessel density of the hippocampal regions (CA1 and CA3) at 5 months and a complete reduction in all areas by 18 months [134]. 

The underlying cause or causes of AD, particularly LOAD, is unknown. In addition, finding a suitable model that spontaneously develops the underlying pathologies remains challenging. However, a long-living (18 years) strictly herbivorous rodent (*Octodon degus*) was found in the upper mountainous regions of Chile [140], which does spontaneously develop cognitive decline with the altered cholinergic transmission, hyperphosphorylated-tau, β-amyloid plaques, NFT, and neuroinflammation in the brain [141,142]. Moreover, said model performs coprophagy (fecal reingestion) and is susceptible to diabetes mellitus, a major risk factor for AD. Regarding drug development, its utility is still in the early stages. Although some data suggested that *O. degus* raised in captivity at lower altitudes cannot reproduce the AD hallmarks reported earlier [143] in a laboratory environment.

## 6. Emerging Models

The National Institute on Aging (NIA) established the model organism development and evolution for late-onset Alzheimer’s disease (MODEL-AD), which addresses the following points: (1) to develop new models of LOAD using humanized Aβ and tau to recapitulate human AD pathology, (2) to identify novel genetic variants that increase the risk of LOAD, and (3) to develop a robust preclinical testing line for testing newer therapeutics [109]. A knock-in model, hAβ-KI has been developed which expresses a humanized Aβ sequence within the murine APP gene. hAβ-KI mice models display age-dependent behavioral abnormalities and synaptic plasticity, but do not develop amyloid plaques [144]. hAβ-KI mice develop age-related insoluble Aβ, decreasing soluble Aβ, suggesting that this shift from soluble to insoluble Aβ may be playing a mechanistic role in the AD progression [145]. hAβ-KI mouse models produce human Aβ at physiological levels in cells which normally express APP, without the addition of any FAD mutations or over-expression of APP or its metabolites [109]. Amyloid plaques were not observed at up to 22 months of age, neither microgliosis nor astrogliosis was observed in 22-month-old mice. The neuron number was the same in the hippocampal region of hAβ-KI and wild-type mouse models. Cognitive impairment was observed at 10 months. Levels of pro-inflammatory cytokines increased in the hAβ-KI mouse model, with a concomitant decrease in the level of anti-inflammatory cytokines [144].

hAβ-KI model displays features like (1) wild type sequence of hAPP cleavage product without any FAD mutations, (2) to produce APP of murine physiological level endogenous gene-regulatory elements were used and (3) the exon encoding Aβ is flanked by *loxP* site to control Aβ/APP production to understand cell-based mechanistic studies [145]. Based on the results of a transcriptomics study (done in 2022) of the brain tissue of hAβ-KI, very few genes were found to be differentially regulated when compared to wild-type mice [146]. A group of researchers in 2021 [145], performed a series of experiments to quantify Aβ level at different ages of mice. The results displayed an increase in insoluble Aβ with a maximum level at around 18–22 months of age, whereas soluble Aβ42 and Aβ40 decreased. This increase in the level of insoluble Aβ was related to the fact that humans and mice Aβ have three different amino acids, which result in boosted amyloidogenicity compared to human sequence [145,147]. The level of proinflammatory cytokines increases and that of anti-inflammatory cytokines decreases in aged hAβ-KI model [148]. Another set of experiments revealed that at physiological level humanized Aβ can cause alterations in gene expressions for processes like metabolism, synaptic plasticity, and memory related functioning [145]. This line of mouse models can be successfully used to introduce and identify factors that may be causing the risk of AD. Genetic susceptibility to LOAD is more complex than EOAD due to variations in genes associated with increased risk of varying degree. A detailed study on the phenotypic characteristics of hAβ-KI showed significant changes in cognition, inflammation, synaptic plasticity, and OC+/PAS granule formation by changing mouse Aβ with human wild-type isoform. However, detailed investigation did not reveal any amyloid aggregation in hAβ-KI brains, concluding that additional factors are required to form amyloid plaques [149]. OC+/PAS granules are associated with astrocytes in brain regions that are relevant to AD. Transcriptomic analysis revealed alterations in approximately 15 genes in the hAβ-KI mice, which are also reported in case of humans [145]. Halting the expression of Aβ in hAβ-KI mice alleviates cognitive impairments. Synaptic deficits are known to occur before amyloid plaques. These studies support that Aβ oligomers are the main toxic species leading to AD [145]. hAβ-KI models can be used to investigate primary risk factors for AD and to study the effect of other multiple risk factors responsible for pathogenesis of disease. 

## 7. Behavioral Studies

The cognitive test is required to assess the ability to learn and think. In contrast, behavioral tests are required to determine an animal’s ability to respond to a stimulus and to determine its sensory responses. As the AD pathology is severe in the entorhinal cortex and the hippocampal region of the medial temporal lobe [150], the medial temporal lobe is responsible for learning, memory, and emotional behavior. The hippocampus, on the other hand, is responsible for spatial memory [151] and gets damaged at the early stages of AD; hence, it is considered essential to study the pathophysiology of the disease [152,153,154]. Therefore, NHP and rodent models are developed to measure hippocampal-dependent memory, which can showcase human memory [27,155,156].

Different types of cognitive-behavioral assays are performed which illustrate similar defects in human AD, namely, fear conditioning (FC) task studies the associative memory, radial arm water maze (RAWM) task studies the short-term memory, affected early in the disease, and Morris water maze (MWM) task studies long term memory, which is affected late in disease [154]. Analyzing animal behavior has become an essential tool in translational neuroscience and for studying the physiological mechanism of a neurological disorder, modifications/changes induced by chemical treatment/genetic manipulations, and the efficacy of novel drugs that reverse the phenotypic conditions in the disease models. These behavioral-cognitive assays in mouse models can provide information regarding the validity and efficacy of the new drugs target/compounds, but the only limitation is that a few cognitive functions are unique to humans and cannot be measured in murine models [154]. 

## 8. Limitations of Murine Models

Using an animal model gives us a clear insight into the pathology of a disease that is difficult to monitor in humans. For studying diseases such as AD or any other neurodegenerative disorders, animal models exhibit pathologies within months compared to years in humans. Using such models has yielded encouraging results, but their clinical value remains uncertain as the mice do not naturally develop neurodegeneration (which is interesting in itself), thus, the focus is mainly on familial AD (FAD) [157] and not on sporadic human AD (SAD). Likewise, there are considerable differences between the immune system of both mice and humans which most models fail to address. The effectiveness of a transgenic animal model in mimicking certain diseases relies on how they respond to behavioral assessments [2]. Until now, murine models were ineffective in exhibiting progressive loss of cholinergic neurons in AD [78]. However, different studies may provide different results using AD mouse models. Some newer models show physiological characteristics like human amyloid protein with quantifiable levels of amyloid plaques but still lack neurodegeneration, the main feature of human AD [158,159,160]. 

Furthermore, there is no consistency in the results when working with AD mouse models. Very few rodent models allow us to study the formation of plaques and tangles simultaneously, as the presence of both is required to diagnose AD and to understand how the interaction between plaques and tangles contributes to the progression of Alzheimer’s [11]. Evidence shows that tau increases Aβ toxicity leading to the understanding that both pathological characteristics are essential to replicate toxicity in human AD [11]. Mice have a shorter life span and a smaller and less developed prefrontal cortex, which can be a significant drawback for studying age-related neurodegenerative diseases, such as AD [157,161]. 

Although transgenic murine models remain the mainstay of AD laboratory research, some earlier models exhibiting overexpression are currently being phased out. The primary reason is that the models demonstrated incomplete pathologies such as widespread Aβ pathology without neuronal loss (strongest AD correlate), cognitive deficits [162], and Tau pathology. Conversely, other models have prioritized the overexpression of human tau proteins at the expense of Aβ pathology. Accordingly, these incomplete pathologies and the limited information provided have led to the emergence of newer models that better emulate the complete pathology of the disease, some of which will be addressed in later sections. 

APP overexpression in transgenic models results in overproduction of APP fragments and Aβ that makes it difficult to differentiate between functional Aβ and other fragments. Behavioral abnormalities may be induced due to APP overexpression before Aβ pathology [33,45,163]. Differences in promoters, transgene constructs and mouse strains makes it difficult to standardize the phenotypes of different models [164,165]. APP knock-in mice can be considered models to study preclinical AD pathology as these do not express tau pathology or neurodegeneration like APP overexpression mice [163]. APP gene in mouse models contain multiple mutations, which is not the case with humans, and as a result may interact with each other and not represent clinical AD precisely [163]. Mouse brains contain only three variants of tau (MAPT gene product) whereas humans have six [163]. Cross breeding of mutant mice may lead to generation of additional phenotypes [67,166] as well as increase premature death (10 weeks) in some lines [49,163].

## 9. Neuro Imaging in Murine Models

In addition to detecting changes in Aβ and tau levels, a definitive AD diagnosis also requires a multi-layered (0.7 mm thickness/per layer) image of a patient’s brain presenting significant NFT and plague deposition post-mortem. This can be achieved using various imaging modalities, including magnetic resonance imaging (MRI), near-infrared fluorescence (NIRF)g and positron emission tomography (PET).

MRI has been used to assess the metabolic profiles of APP, 3xTg-AD and hyperactive 5xFAD models. For example, [167] amyloid plaques as small as 35 μm have been detected in APP mouse models using magnetic resonance microimaging (MRMI) at a high magnetic field (9.4 T). Moreover, age-dependent reductions in metabolite concentrates (acetyl aspartate (NAA), glutamate and glutathione) and elevations in taurine (related to the degree of neuroprotection) [168] in the cerebral cortex of 19-month APP mice analogous to AD patients have been detected using single-voxel H(proton) magnetic resonance spectroscopy ((1)H MRS) at high magnetic field. Furthermore, in vivo H-MRS found a significant reduction and elevation in taurine in 3xTg-AD mice compared to age-matched controls [169]. Moreover, a recent study using 3xTg-AD to evaluate the potential of scFv-h3D6 immunotherapy showed a partial recovery in brain volume reduction of the inflammatory marker, IL-6, compared to the controls at 12 months [170]. 

Magnetic resonance spectroscopy (MRS) has been used to identify biomarkers of AD progression in the hippocampi of 5xFAD mice [171]. H-MRS revealed decreased concentrations of Gamma-Aminobutyric Acid (GABA) at 9 and 10 months in the dorsal hippocampus, and increased concentrations of myo-inositol (Myo). Several features of AD pathology are expressed by 5XFAD mice demonstrating its compatibility with studies reliant on brain imaging and pathogenic amyloid measurement. Studies have shown that cerebral metabolic alterations precede the clinical manifestation of AD symptoms. Distinct patterns of cerebral glucose metabolism also help differentiate AD from other neurodegenerative disease causes [171]. Traditionally, cerebral metabolic rates of glucose have been measured utilizing the imaging tracers 2-[18F] fluoro-2-deoxy-D-glucose (FDG) and (18)F-AV-45 (florbetapir) in conjunction with PET to establish a proxy for neuronal activity, which has also been shown feasible for quantifying Aβ in 5XFAD mice. Moreover, a comprehensive evaluation of 5X FAD including whole brain imaging, autoradiography, blood plasma analysis (cytokines and Aβ species) and numerous behavioral measurements, was reported by Oblak et al. [60] Interestingly, the authors concluded that 5XFAD was better suited to AD studies involving sex-based differences, immune dysfunction, and Aβ deposition than translational studies evaluating the efficacy of novel therapeutics targeting cognitive impairment.

## 10. Evolutionarily Closer Models

The evolutionary distance between murine models and humans can be a limitation. Mammals phylogenetically closer to humans include canine and nonhuman primate models (NHP), which are functionally and neuroanatomically similar exhibiting larger brain sizes and providing larger sample volumes for biomarker studies [157]. Age is the greatest factor in the risk of AD, the course of ageing in canine and nonhuman primates provides more significance for life-span-related changes than that of mice. For example, aged dogs mainly accumulate Aß deposits with a similar amino acid sequence to the human protein which is an advantage over transgenic mouse models. However, Tau pathology, including aggregated intraneuronal topographies and pre-tangles, is very limited in aged dogs [142,172,173]. Aß is also presented in canine’s CSF, and the ratio of Aß42/Aß40 declines with age [142]. However, cerebrovascular changes linked to cognitive decline and mitochondrial dysfunction have been reported in canines [142]. Hence the aged dog model is suitable for studying the neuroprotective effect of drug molecules.

NHP models are beneficial animal models for AD since they share genetic similarities with humans, have a developed prefrontal cortex, and exhibit age-related cognitive deficits [174]. Studies confirm amyloid plaque formation in aged monkeys and great apes [175,176,177]. Hyperphosphorylated tau has been reported in *Macaca mulatta* (rhesus monkey), *Papio anubis* (baboon), and *Saimiri sciureus* (squirrel monkey) [178]. Diffuse Aß plaques and vascular amyloid were observed in the brain of *Pongo pygmaeus* (Orangutan) and *Pan troglodytes* (Chimpanzee) but they lacked NFT [179,180]. 

Many aged NHP species develop amyloid plaques without NFT or neuronal loss [181]. The utility of *Macaca mulatta* and the *Macaca fascicularis* (Philippine monkey) in old-world monkeys has been limited due to inconsistent demonstration of AD pathology [157,182,183]. The new world monkeys (Marmosets, Capuchin monkeys, night monkeys, saki monkeys, and spider monkeys) have proven useful for studying different aspects of human aging [184]. The fast-aging (10–12 years) marmosets are particularly attractive as they have a well-developed platform and toolbox for gene editing, neurophysiological and cognitive studies, and neuropathology studies [185]. The literature proposes that NHPs are better models for non-pathological aging due to the absence of AD-like brain pathology and difference in amyloid biochemistry [157,183]. Animal models related to the amyloid cascade hypothesis are shown in Figure 3.

Amyloid plaque deposits, age-related memory deficits, and the atrophy and loss of cholinergic and monoaminergic neurons are well documented in NHPs [186,187,188]. Aged marmoset monkeys have displayed Aß40 in vascular deposits and Aß42 in plaques associated with swollen neurites but not phosphorylated tau [189]. Aged rhesus monkey have displayed an upregulation of Aß42 in body fluids and brain [188,190]. Currently, new models targeting intracerebral delivery of Aß-plaques and neurofibrillary tangles are being chased. The first experimental transmission of ß-amyloid to NHPs was reported in 1993 by Baker et al. [191]. Lifestyle changes have been shown to decrease the risk of AD. A few studies with diet as a risk factor for AD were also performed with NHPs. One such study involving rhesus monkey on calorie restriction (CR) diet displayed better learning and greater preservation of grey matter in frontal and parietal cortices [188,192]. The CR diet modulated inflammation and lessened the oxidatively damaged proteins compared to age-matched controls [188,193,194]. 

NHPs pose major similarities to the human brain than a mouse. Monkeys and humans share a similar morphology and composition of cell types [195,196]. NHPs are closely related to humans in many aspects like physiology, metabolism, the aging process, and genomic regulation. NHPs are most suitable for determining behavioral abnormalities that occur in ND. NHPs make it easier to study certain phenotypes like depressive behavior or cognitive impairment in ND which is not possible in smaller animals [196,197]. Despite all the positive aspects of using NHP models for research, there are some limitations to it as well. It is technically difficult to scale up the reproducibility of these animals as it takes time for them to reach sexual maturity. Age plays an important role when studying ND, so it can be assumed that these animals will show phenotype and neuropathology when they grow old. The cost of maintaining NHPs throughout their life span thus increases [196]. 

NHP models are currently used for preclinical and clinical trials for the evaluation of neuroprotective approach. Several therapies against β-amyloid (vaccines, antibodies), tau protein (vaccines), γ- and β- secretase (inhibitors or modulators) are being researched. NHPs are used in these studies to determine the safety and efficacy of the treatment [188].

## 11. Fluid Biomarkers

In line with the amyloid cascade theory, early AD diagnostics focused on changes in Aβ40, Aβ42, and total tau(t-tau) levels in cerebrospinal fluid (CSF). However, changes in these markers also occur in other neurological diseases and normal aging. Moreover, 25% of AD patients with mild cognitive impairment (MCI) have normal t-tau levels, while CSF t-tau is elevated in 50% of preclinical AD individuals [198]. Therefore, it has been suggested that elevated tau is due to increase production and neuronal plasticity, whereas elevations at the end stages of the disease can be attributed to dying neurons. 

CSF sampling is invasive, and the physical burden on patients and murine models is high. However, the blood absorbs nearly 500 mL of CSF daily, providing a readily accessible sampling pool. Moreover, blood plasma is obtainable at fractional costs rendering it more suited to mass testing. However, t-tau CSF levels correlate poorly with plasma levels, restricting its usage as a biomarker. In addition to total tau, other biomarkers spanning all three stages of AD include phosphorylated tau (p-tau) and amyloid beta oligomers (AβO’s) (Figure 4). The p-tau isoforms, p-tau threonine 181, p-tau threonine 217, and p-tau threonine 231, plasma levels strongly correlate with CSF values and are considered AD’s early mid stage (Braak 3–4) biomarkers and predictors of Aβ^−^ to Aβ^+^ conversion in MCI patients [199].

Low molecular weight AβO is present at the beginning and later stages of the disease and can be found in blood, plasma, and CSF. Li et al. recently reported a highly specific imaging probe for soluble AβO’s known as AN-SP, a fusion of aminonaphthalene-2-cyanoacrylate (ANCA) with a spiropyran (SP) [200]. Investigations regarding the specificity of ANSP for AβOs performed on 8-month-old APP/PS1 transgenic and wildtype mice showed that AN-SP exhibited excellent colocalization with Aβ oligomer-specific antibodies but not with conformational tau pathology. A summary of the current state of NIR imaging probes specific to Aβ can be found in a recent review by Peng et al. [201] (Figure 4).

Earlier findings have reported reduction in Aβ42 level in CSF and plasma of Tg2576 mice, similar to AD patients [202]. Older mice of AD 3xTg, APPPS1, and APP23 show a decrease in Aβ42 level in CSF as compared to young mice [203,204]. In PDAPP mice models, the level of Aβ42 correlates with abundance of plaque [205], which is inverse to what is found in AD [206]. Tg2576 and APP23 mice accumulated different Aβ species (Aβ40 and Aβ42), unlike AD patients where Aβ42 is the predominant species [67]. No consistent changes were found in the plasma level of Aβ_1–40_ in studies using 3xTg, APPPS1, or APP/PS1 mice. The plasma level of Aβ_1–42_ was also not consistent in 3xTg and APP/PS1 transgenic mice. APPPS1 transgenic mice, however, showed a decrease in the plasma levels of Aβ_1–42,_ and Tg2576 mice showed a decrease in both Aβ_1–40_ and Aβ_1–42_ plasma levels [204]. Another diagnostic biomarker CSF pTau/Aβ_1–42_ ratio can be used for predicting prodromal AD stage [204,207]. Sporadic AD human patients tend to show an increase in CSF levels of T-tau and P-tau compared to healthy individuals. Animal brain homogenates were analyzed, revealing higher levels of T-tau and P-tau in older animals compared to younger ones. A study conducted in 2013 indicated that the T-tau levels in CSF of APPPS1 and APP23 mice increased with age [208]. Plasma levels of both T-tau and P-tau were increased in the 3xTg mice compared to wild-type mice [209]. Detection/measurement of P-tau mainly depends on the type of antibodies pairs that are being used in the experiments since P-tau has many phosphorylation sites and multiple fragments in body fluids [204,210,211]. Neurofilament light (NfL) is a biomarker of neurodegeneration, not specific to AD but its increased levels in CSF can predict the start of neurodegeneration in AD, cognitive decline, and structural changes happening in brain [212]. CSF and plasma levels of NfL have been found to be increased in AD patients [213,214,215]. Bacioglu et al. in 2016 reported NfL levels in two transgenic mice, APPPS1 and P301S-tau mice. NfL levels of CSF increased with age, and plasma level increased accordingly in APPPS1 mice. The CSF and plasma level of Nfl increased with age in tau overexpressed P301S-tau mice [216]. Synaptic proteins such as SNAP25, PSD-95, synaptophysin have been reported to decrease in level in AD brains [217]. SNAP25 has been found to be decreased in human AD brains, but its CSF level is increased in AD patients [217,218]. Level of synaptophysin and PSD-95 have been decreased with age in brains of J20 and APP/PS1 mice as reported by Hong et al. [219]. Glial Fibrillary Acidic Protein (GFAP) is the marker for astrocyte activation. GFAP CSF levels are increased in AD patients. Transgenic mouse models APPPS1 [62], 3xTg [220], and APP/PS1 [221] have shown increased levels of GFAP with age. 

Canines, on the other hand, display age-related Aβ deposition and cognitive decline. Levels of CSF Aβ42 and oAβ are found to decrease at the same rate as observed in human AD [222]. BACE1, the enzyme responsible for amyloidogenesis, shows great potential as a therapeutic target. BACE1 inhibitors can be used to decrease its level in CSF and plasma of Aβ42 and Aβ40 in mice and NHPs [223,224]. A few studies have also reported decreased levels of APPβ in the presence of the BACE1 inhibitor in rhesus monkeys [223]. 

## 12. Conclusions

Animal models provide us the advantage of performing pre-clinical and clinical trials for testing new drugs and therapies, and for reviewing the cognitive testing. These models also provide us with the strategy for development of new diagnostic biomarkers. Since bigger mammalian models like dogs and NHPs develop age-related AD pathology, they can prove more helpful in assessing the efficacy and safety of new therapeutics. The mammalian models developed for AD investigations exhibit various disease characteristics. However, none of these models can replicate all the pathophysiological features of human AD. A few key points need to be considered before selecting the animal model for experiments. The gender of the models may affect not only onset of disease but also its progression. Previous studies have reported that female models of 5xFAD Tg mice tend to display AD phenotype and pathologic characteristics before male models [225,226]. Timings for behavioral testing and tissue sample collection also need to be set in a manner that obtains promising results, since AD is an age-dependent disease [226]. Moreover, many transgenic murine models exhibit genetic drift (allelic drift), resulting in Jackson Laboratory (Jax) and the National Institutes of Health (NIH) sub-strains, thus quality safeguards are necessary to monitor and maintain the genomic validity of the transgenic AD models of interest [227,228].

SAD constitutes 95% of the cases, but the models (Transgenic/NHP/Canine) on which anti-AD drug candidates are tested, utilize FAD partial models. This may account for the failure of drugs in clinical trials as the brain distribution of Aβ and tau is different in SAD and FAD [229]. Additionally, a few models express both NFT and amyloid plaques. The crosstalk between Aβ and NFT significantly increases the toxicity [230]; hence, the models displaying pathological characteristics of both Aβ and tau are important to study these interactions during AD progression and therapeutic intervention. Drugs that lower Aβ levels have been the most desirable for drug development. For example, after many clinical trials, it has been shown that anti-Aβ antibody therapies or secretase inhibitor therapies lower Aβ levels in animals and humans. However, the safety concerns remain [231] and anti-amyloid immunization only arrests disease progression but does not improve cognition [232,233]. Over the years, however, considerable improvement has been made in understanding the pathophysiological mechanisms of AD, as well as developing research technologies (neuroimaging, biomarkers assays) that are adaptable from animals to humans and vice versa. Thus, there is a need to develop next-generation animal models that will more precisely predict the efficacy of treatments, enabling the translation of drugs from the lab to the clinic. 

## Figures and Tables

**Figure 1 cells-12-02459-f001:**
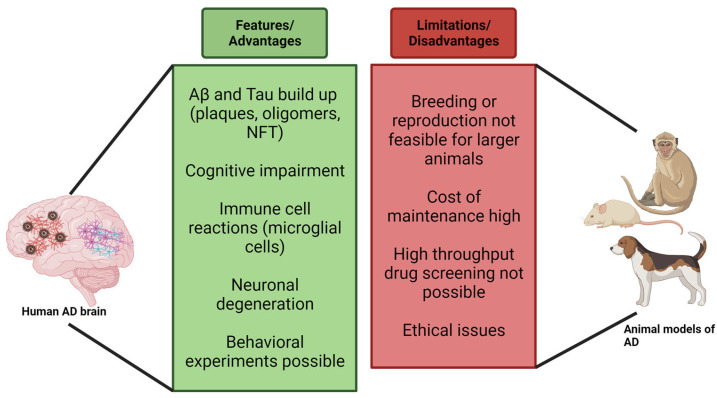
Advantages and disadvantages of using mammalian models to study AD pathology (Created with BioRender). Abbreviations: Aβ (amyloid β); NFT (Neurofibrillary Tangles).

**Figure 2 cells-12-02459-f002:**
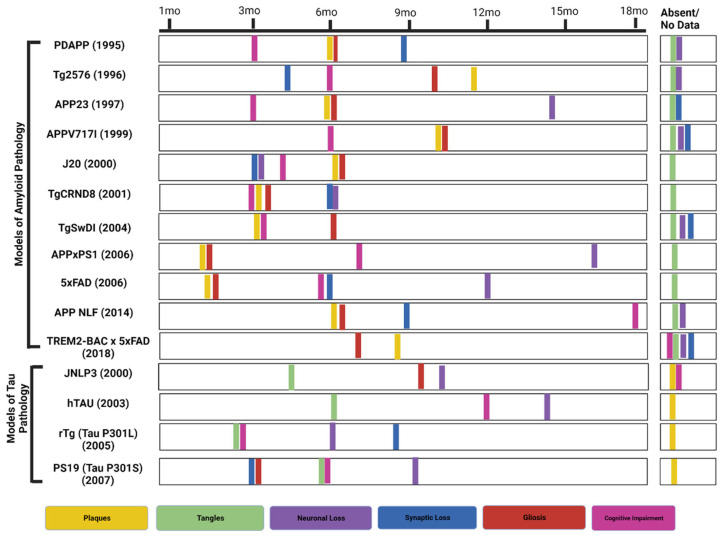
Phenotypic characterization of prominent AD transgenic murine models. (Created with BioRender). Abbreviations: mo (months).

**Figure 3 cells-12-02459-f003:**
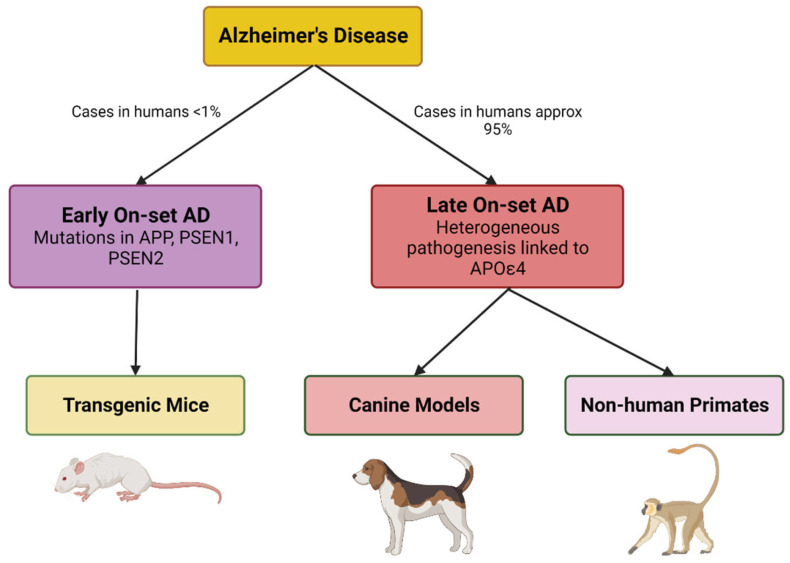
Illustrates the animal models used to replicate Early On-set AD and Late On-set AD (sporadic) pathologies over recent decades. (Created with BioRender).

**Figure 4 cells-12-02459-f004:**
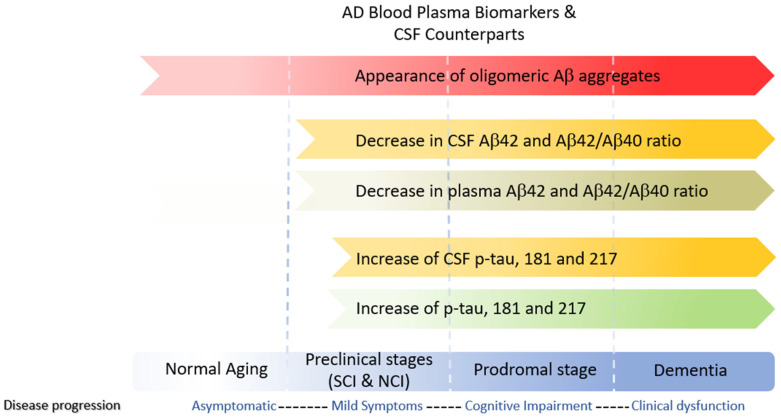
Clinically relevant blood plasma biomarkers used in AD detection and Aβ: Amyloid-beta; CSF: cerebral spinal fluid; NCI: non-cognitive impairment; OAβ: oligomer amyloid beta; SCI: subjective cognitive impairment. Modified with permission [199].

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
