# Peer review of "Mammalian Models in Alzheimer’s Research: An Update"

_cells, 2023, doi:10.3390/cells12202459_

Round 1
Reviewer 1 Report (Previous Reviewer 1)
"17. LOAD Murine Models" in Line 338 seems unnecessary.
Author Response
Dear reviewer,
Thanks for reviewing our manuscript and your valuable comments.
We have revised accordingly as following.
"17. LOAD Murine Models" in Line 338 seems unnecessary.
It was a numbering issue that has been corrected.
Reviewer 2 Report (New Reviewer)
the review is welcome one, and discusses the animal models and the way forward in this preclinical field. The manuscript is not organized well for a smooth reading. lots of data has been given which needs to be rearranged
major points
1 although the topic is good, however, significant information is not provided. As all the information provided is the over view only. molecular mechanisms of the limitations is missing
2 the non human primate models should be covered more extensively. limitations, advantages and roadblocks
3 table 1 and 2 information is very widely available. authors should discuss more about their limiting factors
4. in abstract authors mentioned " application of emerging models (hAβ-KI)". however in manuscript there is only one para of this model with just a overview? what's is the main benefit of this model over others. authors need to focus on that because their review is based on that rationale.
5. review focus on only FAD? authors can clear this in title. Else SAD models may be discussed as well (i.e. models developed with metal exposures or other chemical etc. like reviewed in PMID: 25307560 )
I recommend, adding sections about molecular mechanisms behind currently used animal models of AD. Give brief about which animal models are obsolete. A figure would have been more useful for depicting the Molecular mechanisms.
Future aspects of studying animal models, and why translational studies using these promising animal models fail in clinical setting.
Review is not focussed, first summarize all animal models in one or two tables (like already did by authors, don't add text much about that, mention limitations in table itself. club sections 2 -16 under one heading. even authors can think of clubbing murine models under one section and then non-human primates models in other section and other mammalian models in another section). then focus primarily on currently used and promising animals models.
If authors have taken data of table 1, and 2 from other publication/website, then provide the reference of that article or website.
minor errors
1. line 187, heading 5xFAD. 5 digit is missing and many similar errors
2. too much of typo errors in the manuscript
I am not that expert about English language usage
Author Response
Dear reviewer.
Our sincere thanks for the comments and suggestions of the reviewer. We have incorporated all the suggestions and comments in the revised manuscript highlighted in red font.
major points
1 although the topic is good, however, significant information is not provided. As all the information provided is the over view only. molecular mechanisms of the limitations is missing
The information has been updated.
2 the non-human primate models should be covered more extensively. limitations, advantages and roadblocks
The information has been updated under the section “Evolutionarily Closer models”
3 table 1 and 2 information is very widely available. authors should discuss more about their limiting factors
The table has been updated as suggested
- in abstract authors mentioned " application of emerging models (hAβ-KI)". however in manuscript there is only one para of this model with just a overview? what's is the main benefit of this model over others. authors need to focus on that because their review is based on that rationale.
The suggested corrections have been made
- review focus on only FAD? authors can clear this in title. Else SAD models may be discussed as well (i.e. models developed with metal exposures or other chemical etc. like reviewed in PMID: 25307560 )
Information updated under the section “LOAD murine models”
I recommend, adding sections about molecular mechanisms behind currently used animal models of AD. Give brief about which animal models are obsolete. A figure would have been more useful for depicting the Molecular mechanisms.
Figure 1 has been added depicting the molecular mechanisms involved in AD pathology.
Future aspects of studying animal models, and why translational studies using these promising animal models fail in clinical setting.
Review is not focussed, first summarize all animal models in one or two tables (like already did by authors, don't add text much about that, mention limitations in table itself. club sections 2 -16 under one heading. even authors can think of clubbing murine models under one section and then non-human primates models in other section and other mammalian models in another section). then focus primarily on currently used and promising animals models.
Sections have been divided accordingly and corrections have been made as suggested
If authors have taken data of table 1, and 2 from other publication/website, then provide the reference of that article or website.
References have been checked and updated
minor errors
- line 187, heading 5xFAD. 5 digit is missing and many similar errors
Correction has been done
- too much of typo errors in the manuscript
Corrections have been made
Round 2
Reviewer 2 Report (New Reviewer)
1. the detaisl of hAβ-KI model is not sufficient, which were highly claimed in abstract section. either it can be removed from abstract or explained at molecular level.
2. the manuscript is still not organized. information present in tables and figures can be omitted from text
3. Fig. and table should be provided with Abbrevations full form below the table/fig.
4. conclusion/future directions is not supported by the theme of the article. it is not like that animal models have not contributed to understanding of the AD. in fact it has contributed a lot!!!
Author Response
Our sincere thanks for the comments and suggestions of the reviewer. We have
incorporated all the suggestions and comments in the revised manuscript highlighted in
red font.
- the detaisl of hAβ-KI model is not sufficient, which were highly claimed in abstract section. either it can be removed from abstract or explained at molecular level.
The information has been updated according as per the suggestion.
- the manuscript is still not organized. information present in tables and figures can be omitted from text
Tables have been added under the “supplementary tables” section.
- Fig. and table should be provided with Abbrevations full form below the table/fig.
Abbreviations have been updated as per suggestions.
- conclusion/future directions is not supported by the theme of the article. it is not like that animal models have not contributed to understanding of the AD. in fact it has contributed a lot!!!
The information in the conclusion section has been updated.

This manuscript is a resubmission of an earlier submission. The following is a list of the peer review reports and author responses from that submission.
Round 1
Reviewer 1 Report
The manuscript titled "Animal Models in Alzheimer's Research: An Update" provides a comprehensive overview of the current animal models used in dementia studies. Each model is briefly introduced, and the pros and cons of each model are discussed, making it easy for readers to understand and use the models. However, the manuscript requires some revisions to improve its clarity and readability. Additionally, there are several typos throughout the manuscript that need to be corrected. Overall, with these revisions, the manuscript would be suitable for publication in the journal.
1. line 12 single aspects
2. line 13-18 the sentence too long. Divide it into two or three sentences, please.
3. line 18-19 “animal-human behavioral translation” – a more detailed explanation needed
4. line 53 Ab -- in full needed since it comes first
5. line 64 twenty-plus - Does it mean more than 20?
6. line 90-91 neurofibrillary tangles (NFT) -- already appeared in the earlier section
7. line 93 brain accumulation – accumulation in brain
8. line 121 LTP/LTD -- in full needed since it comes first
9. line 142 synomyous
10. line 144 upto
11. line 159 [47] move it to an appropriate place, such as after “stress occur”
12. line 175 this 1999 model – revision needed
13. line 181 beta-amyloid (Aβ)
14. line 181-182 cerebral amyloid angiopathy (CAA)
15. line 197 Aβ amyloid plaques – redundancy
16. line 208 [61] move it to the end of the sentence
17. line 217-218 what is the “said mutation”?
18. line 218-219 at years end – revision needed
19. line 223 [64] move it to the end of the sentence
20. line 241 “…expressed by microglia in the Brain…” – expressed in microglia in brain
21. line 247-248 move the sentence in an independent paragraph and complete the sentence
22. Table 1 and 2 hope you create a section for the reference
23. line 253 what does it mean “4R”?
24. line 290 [87] move it to the end of the sentence
25. line 298 Ab pathology
26. line 299-300 move the sentence in an independent paragraph
27. line 319 remove “
28. line 336, 347 what it the “said model”?
29. line 344, 386-387 Octodon degus in italic
30. line 358 [102] a period needed
31. line 375 remove “:”
32. line 394 the cynomogolous -- the article not needed
33. line 398 Marmosets – marmosets
34. line 408-425 name of a gene should be written in italic
35. line 427 The NIA – describe it in full
36. line 452 the otherhand – the other hand,
37. line 474-475 “translatable drug development association to date” it is difficult to understand its meaning.
38. line 479 no. line 482 nos. revise these two
39. line 478-483 the sentence too long. Please divide it into two or three sentences.
40. line 481 AD Seeing – AD seeing
41. line 498-502 the sentence too long. Please divide it into two or three sentences.
Reviewer 2 Report
The paper of Sharma and colleagues is a one more (short) review on animal models of Alzheimer’s disease (AD). Due to the growing efforts in modeling AD and to the concomitant low translational potential of the engineered models, mainly observed in drug testing (therapeutics validated in animal models but with quasi-constant failures or mitigated results in clinical trials), such review papers might be of importance to understand the gap between animals-patients and decipher the limits of actual models.
The present paper however does not fulfill important criteria for publication, for different reasons.
>There is an overall lack of originality in the data compilation: It is clearly not obvious what this paper brings in terms of novelty (new emerging models? new hypothesis concerning models validity / translational potential? etc.). Compared to the tons of reviews on this topic, much of provided information in this paper, has been already exposed, often in a more comprehensive and structured manner, in numerous other review articles.
>The overall plan of the review is curious and fuzzy.
Examples:
-The review focuses on animal models but the abstract mentions “microfluidics applications” (link with animal models?), not detailed later in the paper (!).
-Section “8. Other transgenic models”: models described are non-transgenic (spontaneous aging of dogs, NHPs)…
>More importantly, there are some disputable, partial or even wrong information provided in this review paper.
-“Pathologically pure AD is characterized by […] intracellular neurofibrillary tangles (NFT) and hyperphosphorylated tau protein deposits”: Actually NFTs are made of hyperphosphorylated tau.
-“LOAD is synonymous with the carriage of the apolipoprotein E (APOE) ε4 allele”: Wrong!
-Page 2: the authors do not make the link between BACE1 and ß-secretase…
-“additional mutations in the gene for presenilin 1 (PS1) in chromosome 14 and presenilin 2 (PS2) in chromosome 1 observed in AD families [30] has allowed the observation of additional pathologies [in animal models]”: the most important impact of adding PS1/2 mutated transgenes to APP transgene is to accelerate the onset of pathology, not to modify its nature.
-The amyloid cascade has not been “established through a series of findings between 1984 and 1987” but later mainly in relation with the discovery of the 1st genes mutations linked to FAD…
- “The first amyloid model was presented in 1995 [36], that expressed high levels of human APP”: true but this is hAPP with a mutation (V717F)!
-Tg2576 “can sometimes display hyperphosphorylated tau tangles”: Wrong! Abnormal tau phosphorylation is observed in peri-plaques neurites but no morphologically-defined tangles are observed in APP mice models. The cited paper (Puzzo et al.) mentions accumulation of hyperphosphorylated tau in old Tg2576 but this is not under the form of tangles.
-Similarly the terms “tauopathy” to describe the APP23 mice is overstated.
-The short review of the 5xFAD model lacks critical information concerning this mouse line: presence of intracytoplasmic Aß (or APP CTFs) strongly associated with cell toxicity-loss. Other models presenting similar phenotypes (eg APPxPS1-Ki studied by Thomas Bayer) should be cited.
-TREM2 Humanized (R47H) x 5XFAD model: for the reader it is not understandable what is the rationale and interest of this model as no explanations are given on the R47H variant in humans…
-The selected list of “amyloid models” is largely incomplete and indeed cannot be exhaustive. But it remains necessary to include standard APPxPS1 models which have been very instrumental in preclinical research (eg APPxPS1dE9 from Jankowsky). No information are provided in the present review on APPxPS1 models.
-Also the more recent models based on KI (eg of APP) engineered at the Riken by Saito-Saido are not described or evoked in the review.
-The rTg4510 models has been recently criticized (co-expression of additional genes) by Gamache et al (2019). This is imperative to present the limits of the cited models.
-The choice of the “tau models” is also curious. One of the most used models, the PS19 line (V. Lee) is not even cited in the present review.
-The 3xTg line of LaFerla is not a tau model but a composite model developing both Aß and tau pathologies.
-“ Mice have a shorter life span and a smaller and less developed prefrontal cortex which can be a significant drawback for studying age-related neurodegenerative diseases like AD”: somehow trivial and questionable (rationale of focusing on prefrontal cortex?).
-“NFTs are absent in canines and nonhuman primate species. In aged dogs, however, hyperphosphorylated tau has been reported”: once again there is a confusion in terminology and erroneous statements. Aged dogs mainly accumulate Aß deposits. Tau pathology, including aggregated intraneuronal topographies and pretangles, is observed spontaneously in numerous species (NHPs, cats, polar bears etc.) but is very limited in aged dogs.
-“Analysing animal behaviour has become an important tool in areas of translational neuroscience and for studying physiological mechanism of a neurological disorder”: also obvious and trivial.
-“cognitive functions are unique to humans and cannot be measured in an animal model (for example, WM tasks that are related to language and math)”: also very trivial statement.
>Formal aspects (English writing) should be revised. Several sentences are difficult to understand or awkwardly written. Some examples:
-“models targeting single aspects of AD pathogenesis presenting at various stages of AD have failed”: not understood.
-“such deficits are already prevalent in many systems”. What do the authors mean by systems?
-“peripheral memory”: interesting concept. What it is?
-“the AD pathology varies with the age at which each animal develops certain features”. Which features? Not understandable.
-“hAβ-KI mice models display age-dependant [sic] impairments in behavioural abnormalities”. What is an impairment of an abnormal behavior?
>Illustrations
-Figure 1 is useless. Several abbreviations are note explicited (EP? ALP?).
-Figure 2 is a copy/paste of the Alzforum website (animal models database).